# Long-Term Prognosis of Patients with Myocardial Infarction Type 1 and Type 2 with and without Involvement of Coronary Vasospasm

**DOI:** 10.3390/jcm9061686

**Published:** 2020-06-02

**Authors:** Ryota Sato, Kenji Sakamoto, Koichi Kaikita, Kenichi Tsujita, Koichi Nakao, Yukio Ozaki, Kazuo Kimura, Junya Ako, Teruo Noguchi, Satoshi Yasuda, Satoru Suwa, Kazuteru Fujimoto, Yasuharu Nakama, Takashi Morita, Wataru Shimizu, Yoshihiko Saito, Atsushi Hirohata, Yasuhiro Morita, Teruo Inoue, Atsunori Okamura, Toshiaki Mano, Kazuhito Hirata, Kengo Tanabe, Yoshisato Shibata, Mafumi Owa, Hiroshi Funayama, Nobuaki Kokubu, Ken Kozuma, Shirou Uemura, Tetsuya Toubaru, Keijirou Saku, Shigeru Ohshima, Kunihiro Nishimura, Yoshihiro Miyamoto, Hisao Ogawa, Masaharu Ishihara

**Affiliations:** 1Department of Cardiovascular Medicine, Graduate School of Medical Sciences, Kumamoto University, Kumamoto 860-8556, Japan; ryout11@kumamoto-u.ac.jp (R.S.); kaikitak@kumamoto-u.ac.jp (K.K.); tsujita@kumamoto-u.ac.jp (K.T.); 2Division of Cardiology, Saiseikai Kumamoto Hospital Cardiovascular Center, Kumamoto 861-4193, Japan; koichi-nakao@saiseikaikumamoto.jp; 3Department of Cardiology, Fujita Health University, Toyoake 470-1101, Japan; ozakiyuk@fujita-hu.ac.jp; 4Cardiovascular Center, Yokohama City University Medical Center, Yokohama 236-0004, Japan; c_kimura@yokohama-cu.ac.jp; 5Department of Cardiovascular Medicine, Kitasato University School of Medicine, Sagamihara 252-0375, Japan; jako@jichi.ac.jp; 6Department of Cardiovascular Medicine, National Cerebral and Cardiovascular Center, Suita 565-8565, Japan; tnoguchi@ncvc.go.jp (T.N.); yasuda.satoshi.hp@ncvc.go.jp (S.Y.); ogawah@ncvc.go.jp (H.O.); 7Department of Cardiology, Juntendo University Shizuoka Hospital, Izunokuni 410-2295, Japan; suwatetsu3000@yahoo.co.jp; 8Department of Cardiology, National Hospital Organization Kumamoto Medical Center, Kumamoto 860-0008, Japan; fujimoto@kumamed.jp; 9Department of Cardiology, Hiroshima City Hospital, Hiroshima 730-8518, Japan; modified-bruce@tenor.ocn.ne.jp; 10Division of Cardiology, Osaka General Medical Center, Osaka 558-8558, Japan; tmorita@sc4.so-net.ne.jp; 11Department of Cardiovascular Medicine, Nippon Medical School, Tokyo 113-8603, Japan; wshimizu@nms.ac.jp; 12First Department of Internal Medicine, Nara Medical University, Kashihara 634-8521, Japan; yssaito@naramed-u.ac.jp; 13Department of Cardiovascular Medicine, The Sakakibara Heart Institute of Okayama, Okayama 700-0804, Japan; hirohata@tg7.so-net.ne.jp; 14Department of Cardiology, Ogaki Municipal Hospital, Ogaki 503-8502, Japan; ysmorita@med.nagoya-u.ac.jp; 15Department of Cardiovascular Medicine, Dokkyo Medical University, Tochigi 321-0293, Japan; inouet@dokkyomed.ac.jp; 16Department of Cardiology, Sakurabashi Watanabe Hospital, Osaka 530-0001, Japan; a_okamura@watanabe-hsp.or.jp; 17Institute for Clinical Research, Kansai Rosai Hospital, Osaka 660-8511, Japan; mano-toshiaki@kansaih.johas.go.jp; 18Department of Cardiology, Okinawa Chubu Hospital, Uruma 904-2293, Japan; kheart911@yahoo.co.jp; 19Division of Cardiology, Mitsui Memorial Hospital, Tokyo 101-8043, Japan; kengo-t@zd5.so-net.ne.jp; 20Department of Cardiology, Miyazaki Medical Association Hospital, Miyazaki 880-0834, Japan; yshibata@cure.or.jp; 21Department of Cardiovascular Medicine, Suwa Red Cross Hospital, Suwa 392-8510, Japan; mafumi-oowa@suwa.jrc.or.jp; 22Division of Cardiovascular Medicine, Saitama Medical Center Jichi Medical University, Saitama 330-8503, Japan; funahiro@omiya.jichi.ac.jp; 23Department of Cardiovascular, Renal and Metabolic Medicine, Sapporo Medical University, Sapporo 060-8543, Japan; kokubu@sapmed.ac.jp; 24Department of Cardiology, Teikyo University, Tokyo 173-8606, Japan; PXE00364@nifty.com; 25Department of Cardiology, Kawasaki Medical School, Kurashiki 701-0192, Japan; suemura@med.kawasaki-m.ac.jp; 26Department of Cardiology, Sakakibara Heart Institute, Tokyo 183-0003, Japan; ttobaru@shi.heart.or.jp; 27Department of Cardiology, Fukuoka University School of Medicine, Fukuoka 814-0133, Japan; saku-k@fukuoka-u.ac.jp; 28Department of Cardiology, Gunma Prefectural Cardiovascular Center, Maebashi 371-0004, Japan; ohshima.s@cvc.pref.gunma.jp; 29Department of Preventive Cardiology, National Cerebral and Cardiovascular Center, Suita 565-8565, Japan; knishimu@ncvc.go.jp (K.N.); miyamoty@ncvc.go.jp (Y.M.); 30Department of Cardiovascular and Renal Medicine, Hyogo College of Medicine, Nishinomiya 663-8501, Japan; ishifami@fb3.so-net.ne.jp

**Keywords:** acute myocardial infarction, clinical outcomes, coronary vasospasm, universal definition of myocardial infarction

## Abstract

While prognoses in relation to myocardial infarction (MI) type have been elucidated in past reports, the results were not consistent, perhaps due to occurrence of Type 2 MI with CVS and its mortality. The Japanese registry of acute Myocardial Infarction diagnosed by Universal Definition (J-MINUET) is a prospective multicenter registry in Japan. In contrast to thromboembolic event-related Type 1 myocardial infarction (MI), clinical features of Type 2 MI, including coronary vasospasm (CVS), are varied due to the heterogeneous nature of its development. To elucidate the MI type-related all-cause mortality, 2989 consecutive patients with AMI were stratified as Type 1 MI, Type 2 MI with CVS, and Type 2 MI with non-CVS. Most patients (*n* = 2834; 94.8%) were classified as Type 1 MI and 155 patients (5.2%) were classified as Type 2 MI. Of the Type 2 MI patients, 87 (56% of Type 2 MI) were diagnosed as MI with CVS. Although the 3-year mortality was comparable between Type 1 and Type 2 MI patients, significant differences were observed between Type 2 MI with CVS and with non-CVS (3.4% and 22.1%, *p* < 0.001). Among Japanese patients with AMI, mortality rates between Type 1 MI and Type 2 MI are comparable, but further stratification of Type 2 MI (with or without CVS) may be useful in predicting the prognosis of patients with Type 2 MI.

## 1. Introduction

The principle of the ESC/ACC document published in 2000 regarding the redefinition of myocardial infarction (MI) was that myocardial damage detected by abnormal cardiac biomarkers [preferably cardiac troponin (cTn)] during acute myocardial ischemia should be labeled as MI [1]. The second universal definition of MI (UDMI) issued by the European Society of Cardiology (ESC)/American College of Cardiology Foundation (ACCF)/American Heart Association (AHA)/World Heart Federation (WHF) Task Force in 2007 introduced a new classification of MI, specifying 5 subcategories [2]. After the third UDMI in 2012, included a correction related to patients undergoing coronary interventions [3], the fourth UDMI in 2018 was required for several reasons, including the predominance of high sensitivity cTn assays. Although the clinical criteria for MI have not changed, these assays have significantly altered the way in which MI is evaluated utilizing serial change of cTn [4,5,6]. However, the classification of UDMI based on high-sensitivity troponin T has not necessarily led to improved clinical outcomes, and more detailed classification and treatment approaches based on individual conditions have been suggested [7,8].

In contrast to Type 1 MI, which has been defined as spontaneous MI related to ischemia due to a primary coronary event, the causes of Type 2 MI, secondary to ischemia due to either increased oxygen demand or decreased supply, are diverse, and its clinical features are variable due to the heterogeneous mechanisms of onset. Several clinical trials have reported that patients with Type 2 MI have a poorer prognosis than patients with Type 1 MI [9,10,11,12].

Coronary vasospasm (CVS) is one of the causes of MI, including Type 2 MI. However, none of these studies examined the prognostic impact of CVS. Generally, the prognosis for patients with CVS is good, with less than 5% of cases developing acute MI (AMI) [13]. Nonetheless, CVS can also cause sudden death [14]. A previous study has shown that coronary spasm is more frequent among Japanese patients than among Caucasian patients [15]. Thus, it has been suggested that CVS is involved in the onset of AMI, especially among patients of Asian descent. As such, we sought to compare the incidence rate and long-term prognosis for Japanese patients with Type 1 and Type 2 AMI. We hypothesized that prognosis may vary among patients with the heterogeneous clinical features of Type 2 MI, especially in patients with CVS-related Type 2 MI. Further stratification of Type 2 MI patients by presence or absence of CVS and subsequent related clinical outcomes may offer insights for improvement in patient care.

## 2. Experimental Section

### 2.1. Ethics Statement

All procedures in this study were conducted in accordance with the Declaration of Helsinki and its amendments. The study protocol was approved by each participating institution’s ethics committee. All subjects gave their informed consent prior to study enrollment. Registration was done in the University Hospital Medical Information Network (UMIN)-CTR (http://www.umin.ac.jp/ctr/). Identifier: J-MINUET (UMIN000010037).

### 2.2. Study Population

The Japanese registry of acute Myocardial Infarction diagnosed by the universal definition (J-MINUET) criteria is a prospective, observational, multicenter study conducted at 28 Japanese medical institutions [16]. Consecutive patients hospitalized within 48 hours of MI onset between July 2012 and March 2014 were enrolled in the present study. The study protocol has been reported previously [16]. In brief, AMI was diagnosed using the third universal definition, in which cardiac troponin is the preferred biomarker of myocardial injury. This registry contains only Type 1 AMI (spontaneous MI related to ischemia from primary coronary event) and Type 2 AMI (MI secondary to ischemia because of either increased oxygen demand or decreased supply). The selection of cardiac troponin T (cTnT) or cardiac troponin I (cTnI) as the biomarker was left to operator discretion with the cutoff value reflective of that specific facility. cTn measurements are not normally required for patients whose creatine phosphokinase (CPK) levels are more than twice the normal upper reference limit (URL). The frequency of cTn and CPK measurements and the time interval data were not prespecified and remained at the discretion of individual clinicians. ST-segment elevation MI (STEMI) was diagnosed in the presence of new ST elevation at the J point in at least 2 contiguous leads. New or presumably new left bundle branch block was considered a STEMI equivalent. To elucidate the impact of CVS-associated Type 2 MI, we stratified 2989 consecutive MI patients into Type 1 MI, Type 2 MI with CVS, and Type 2 MI with non-CVS (Figure 1).

### 2.3. Clinical Outcomes and Definitions

The primary endpoint was 3-year all-cause mortality. Treatment and in-hospital clinical event data were collected at the time of hospital discharge. Clinical follow-up after index MI procedure was performed through a review of medical records, telephone contacts, and mailed questionnaires.

A diagnosis of MI was based on the ESC/ACCF/AHA/WHF Task Force guidelines for the third Universal Definition of Myocardial Infarction published in 2012 [3]. Type 1 MI was defined as the detection of a rise and/or fall in cTn values, with at least one value above the 99th percentile of the URL, and with at least one of the following: symptoms of AMI: new ischemic ECG changes; development of pathological Q waves; imaging evidence of new loss of viable myocardium or new regional wall motion abnormality in a pattern consistent with an ischemic etiology; and the identification of a coronary thrombus by angiography, including intracoronary imaging or by autopsy [3]. Type 2 MI was defined as the detection of a rise and/or fall in cTn values, with at least one value above the 99th percentile of the URL, and evidence of an imbalance between myocardial oxygen supply and demand unrelated to coronary thrombosis, and with at least one of the following: presence of symptoms of AMI, new echocardiographic evidence of ischemia, development of pathological Q waves, imaging evidence of new loss of viable myocardium, or new regional wall motion abnormality in a pattern consistent with an ischemic etiology [1]. Diagnosis of CVS was based on ergonovine or acetylcholine provocation tests in acute or chronic phases. Patients whose coronary obstruction was relieved after the injection of nitrate into the coronary artery were diagnosed as Type 2 MI with CVS. Patients with CVS and measurable elevations in cTn but no significant stenosis at the time of coronary angiography were also diagnosed as Type 2 MI with CVS. The standard method of the provocation test are as follows: (1) Acetylcholine provocation test: inject 20, 50, or 100 μg of acetylcholine in solution in 37 °C physiological saline into the coronary artery over a period of 20 seconds (limited with 20 or 50 μg to the right coronary artery). Perform coronary angiography 1 minute after the start of each injection; or, (2) Ergonovine provocation test: inject 20–60 μg of ergonovine in solution in 37 °C physiological saline into the coronary artery over a period of several minutes (about 2–5 minutes). Perform coronary angiography 1–2 minutes after completion of the injection. In the event of an ischemic change on the ECG or chest symptoms, perform angiography at the time of its onset. Coronary spasm during the provocation test was defined as “transient, total, or sub-total occlusion (>90% stenosis) of a coronary artery with signs of myocardial ischemia (anginal pain and/or ischemic ST changes)”.

### 2.4. Statistical Analysis

Continuous variables were expressed as the mean ± standard deviation (SD), and categorical variables were expressed as frequencies and percentages. Continuous and categorical variables between the 2 groups were compared using the Student’s t-test and Chi-square test. Where *p* < 0.05, Bonferroni post-hoc multiple comparisons were performed. Estimates of the cumulative event rates were calculated by the Kaplan–Meier method. Cox proportional hazards regression was used to calculate hazard ratios (HRs) and 95% confidence intervals (CI) as estimates for each endpoint. HRs were adjusted for baseline, angiographic, and procedural characteristics according to the univariate analysis for each event. All variables with *p* < 0.10 in the univariate analysis were considered in the multivariate analysis. Differences with *p* < 0.05 were regarded as statistically significant. To assess the impact of with or without CVS in patients with Type 2 MI after adjustment, relative survival was compared by use of Cox proportional hazards with forced entry method (age, sex, and BMI) with Type 2 MI with CVS as the reference.

In this study, main analyses were performed over a 3-year follow-up period, but the follow-up rate was not sufficient (26.5%); therefore, survival analyses with 1-year follow-up (follow up rate was 84.9%) were also performed as a sensitivity analysis. In addition, we performed an analysis that excluded patients diagnosed by CPK alone over a 3-year follow-up as for confirming the robustness of the primary results. Statistical analyses were performed using SPSS version 24 (SPSS; Chicago, IL, USA).

## 3. Results

### 3.1. Patient Background

Of the 2989 patients enrolled, 2834 were classified as Type 1 MI and 155 as Type 2 MI, with 68 patients classified as Type 2 MI with non-CVS and 87 patients as Type 2 MI with CVS. Among patients with Type 1 MI, 456 (16.1%) were diagnosed with CPK alone at more than twice the upper limit of normal (ULN), and among patients with Type 2 MI, 4 (2.5%) were diagnosed in the same manner. The clinical characteristics of these patients are shown in Table 1.

#### 3.1.1. Type 1 vs. Type 2

Between the 2 MI types, patients with Type 1 MI were older, fewer females, and had a significantly higher Killip classification. In terms of coronary risk factors, the incidence of diabetes mellitus (DM) was higher whereas dyslipidemia was lower in the Type 1 MI group. There was no significant difference between the two groups in the presence of chronic kidney disease (CKD) and current smoking. Serologic studies revealed that maximum CPK level, Hemoglobin, and LDL-cholesterol levels were higher in the Type 1 MI group. Procedurally, urgent revascularization was higher and incidence of non-STEMI (NSTEMI) was lower in the Type 1 MI group.

#### 3.1.2. Type 2 with CVS vs. with Non-CVS

Regarding patients with Type 2 MI with non-CVS and with CVS, Type 2 MI with non-CVS were older, fewer females, had higher heart rates (HR) upon admission, and higher Killip classification than Type 2 MI with CVS group. Regarding coronary risk factors, the proportion of DM patients was lower in the Type 2 MI with CVS group whereas the proportion of CKD patients and dyslipidemia were higher in the Type 2 MI with non-CVS group. While the presence of current smokers was higher in the Type 2 MI with CVS group, history of prior MI, percutaneous coronary intervention (PCI), comorbid atrial fibrillation, prior stroke and peripheral artery disease (PAD) were higher in the Type 2 MI with non-CVS group. Maximum CPK level was higher and Hemoglobin and LDL-cholesterol levels were lower in the Type 2 MI with non-CVS group. Procedurally, urgent revascularization was higher and incidence of non-STEMI (NSTEMI) lower in the Type 2 MI with non-CVS group.

Table 2 shows the medications prescribed at discharge. When Type 1 MI and Type 2 MI groups were compared, usage rates of aspirin, double antiplatelet therapy, proton pump inhibitors (PPIs), β-blockers, angiotensin-converting enzyme inhibitors (ACEIs), hypoglycemic agents, and statins were higher in the Type 1 MI group, whereas calcium channel blockers (CCBs), nitrates, and the use of nicorandil were higher in the Type 2 MI. When Type 2 MI with CVS vs with non-CVS, usage rates of aspirin, double antiplatelet therapy, proton pump inhibitors (PPIs), β-blockers, angiotensin-converting enzyme inhibitors (ACEIs), hypoglycemic agents and statins were higher in the Type 2 MI with non-CVS group, whereas calcium channel blockers (CCBs), nitrates, and the use of nicorandil were higher in the Type 2 MI with CVS group.

### 3.2. Clinical Outcomes

Among the 2989 enrolled AMI patients who met the inclusion criteria, 2537 patients (84.9%) completed one-year follow-up and 793 patients (26.5%) completed 3-year follow-up. Mean follow-up period was 665 ± 368 days and median follow-up time was 765 days. The primary endpoint was comparable between Type 1 MI and Type 2 MI groups (11.9% and 11.6%, respectively, *p* = 0.907). When Type 2 MI was divided into 2 groups, there was a significant difference in the cumulative incidence rate of the primary endpoint among the 3 groups, with the highest rate observed for Type 2 MI with non-CVS group (11.9%, 22.1%, and 3.4% for Type 1, Type 2 non-CVS, and Type 2 CVS, respectively; *p* < 0.001).

When cardiovascular mortality rates were compared, Type 1 MI (*n* = 2834) group had 195 patients (6.8%) while Type 2 MI (*n* = 155) group had 10 patients (6.4%). Of note, 9 of the 10 cardiovascular deaths were patients with Type 2 with non-CVS. Thus, mortality with cardiovascular death in Type 2 MI with non-CVS was 13.2% (9 of 68 patients) and that of Type 2 with CVS was 1.2% (1 of 87 patients).

Kaplan–Meier analysis (Figure 2) showed almost parallel curves during the observational period (Figure 2A). Nonetheless, there were significant differences between the CVS Type 2 MI categories, especially in terms of their divergence in the very acute phase (Figure 2B).

Table 3 and Table 4 show the predictors for the primary endpoint by each MI type. In contrast to Type 1 MI (Table 3, which includes higher age, severe Killip classification, CKD, prior PCI, PAD, and multivessel disease, only urgent revascularization and multivessel disease were significant independent predictors among patients with Type 2 MI (Table 4). Interestingly, urgent revascularization need was not an independent predictor among patients with Type 1 MI.

Table 5 shows the result of Cox proportional hazards regression analysis in patients with Type 2 MI. When the Type 2 MI with CVS was used as the reference, a statistically significantly worse prognosis was found for Type 2 MI with Non-CVS (HR = 5.007, 95%CI 1.338–18.741, *p* = 0.017), even when the analysis was adjusted for age, sex, and BMI with forced entry method.

### 3.3. Sensitivity Analysis

In this study, 793 patients (26.5%) completed 3-year follow-up. There is a possibility that the low follow-up rate of the registered patients may have caused some bias. To confirm the robustness of the primary results, additional survival analyses at 1-year follow-up (follow up rate was 84.9%) were performed. The consistency of findings, compared with 3-year analysis, were confirmed on Kaplan–Meyer analyses (Figure 3). In addition, in this study, some patients were diagnosed with MI by CPK alone. This does not represent the recent clinical standard, so we have performed another sensitivity analysis with excluding patients diagnosed by CPK alone over a 3-year follow-up period. In this patient population (Type 1 MI; *n* = 2377, Type 2 MI; *n* = 151), comparable findings with the results of overall patient population were observed (Appendix A).

## 4. Discussion

The main finding of the present study was that similar mortality rates were observed between Type 1 MI and Type 2 MI in Japanese patients. This finding contrasts with reports from Western countries. Another important finding was the significantly lower frequency of Type 2 MI compared with Type 1 MI in this Japanese registry of consecutive AMI patients. In addition, among the Type 2 MI patients, half presented with CVS and had significantly lower mortality rates than those in the non-CVS group. Finally, it is also important to note there were significant differences in the medications prescribed at discharge across the 3 groups.

Several studies have reported that Western patients with Type 2 MI have a worse prognosis than patients with Type 1 MI [9,10,11,12], yet prognosis was comparable among our Japanese cohort of consecutive patients. It is possible that the frequency of Type 2 MI with CVS is higher in Japanese patients than in Western patients [15]. The Japanese Circulation Society guidelines [17] recommend the use of a drug-induced coronary spasm provocation test for patients who are suspected of having coronary vasospastic angina, as this is difficult to diagnose using non-invasive tests. This may explain differences in diagnosis frequency, with the use of provocation tests likely to lead to improved diagnostic accuracy and the selection of optimal medication for the treatment of coronary vasospastic angina.

Regarding the proportion of Type 2 MI, a significantly lower frequency of Type 2 MI compared with Type 1 MI was observed in this study. A multicenter study from 5 European countries, APACE, showed that the update of UDMI increased the number of newly diagnosed Type 2 MI patients reclassified in UDMI 2012 by 114% (comparison with UDMI 2007) in the same MI patient group [18]. A community-based study in the United States found that a dramatic reduction in Type 1 MI from 2003 to 2012 resulted in similar incidence of Type 1 and Type 2 MI [19]. Even in a recent investigation restricted to adults aged 50 years and younger, the ratio of Type 2 MI was as high as 32% (the actual proportion was 37%, if the patients with myocardial injury were excluded) [20]. Epidemiologic studies in the United States consistently report a higher incidence of NSTEMI than STEMI [21], whereas past studies from Japan have reported a lower incidence of NSTEMI (12% to 31% of AMI) [16,22,23,24]. NSTEMI accounts for the majority of Type 2 MI, so the lower proportion of Type 2 MI in this Japanese patient population may reflect ethnic differences. Furthermore, because this study focused on coronary artery disease, patients with myocardial injury, such as Takotsubo cardiomyopathy or pulmonary embolism, were intentionally excluded at the time of entry. This may have affected the lower proportion of Type 2 MI compared with previous observational studies [19,20,25].

Type 1 MI, which is more common, is a spontaneous form of MI due to arteriosclerotic plaque rupture, and its pathoetiology is continuing to be explored using PCI and appropriate treatment. Primary PCI is performed in various facilities in Japan as compared to Western countries, and PCI is usually selected as the first line of treatment for AMI. Compared to thrombolytic therapy, primary PCI has a higher reperfusion rate, reduced frequency of cardiac events, such as angina pectoris post-MI, earlier discharge rates, and improved prognosis [26,27]. In addition, studies have established the efficacy of pharmacological therapy, with the continued administration of aspirin able to reduce cardiovascular events both in the short and long term [28,29], and the use of statins as a secondary prevention able to lower LDL-cholesterol levels for patients with MI [30,31,32].

Aside from the spontaneous development of MI with obstructive coronary arteries in Type 1 MI, there are various other causes of AMI. MI with non-obstructive coronary arteries (MINOCA) is clinically defined by the presence of the universal criteria of AMI, the exclusion of obstructive coronary artery disease (CAD), and no overt cause for the clinical presentation at the time of coronary angiography. MINOCA has been reported to have a prevalence of about 6% of all MI patients, with no clear clinical features as compared with obstructive CAD, with the possibility of multiple causative comorbid diseases [33]. Such causes include coronary vasospastic angina, coronary embolism, arteritis, such as Takayasu disease or Kawasaki disease, trauma, congenital coronary artery malformations, cardiac hypertrophy, aortic stenosis, metabolic disease, such as thyroid crisis, shock, and coagulopathy [34]. In the present study, we excluded Takotsubo cardiomyopathy, also known as stress cardiomyopathy, and pulmonary embolism, both of which belong to the MINOCA category. MINOCA was included in Type 2 MI, however, it cannot be strictly classified as CVS-related or non-CVS-related. Although secondary prevention of MINOCA with optimal medication has been reported to improve long-term prognosis [35], β2blockers can exacerbate the condition when it is associated with CVS. Therefore, MINOCA should not be treated according to a definite diagnosis but considered as a working diagnosis requiring further evaluation.

The prognosis of Type 2 MI depends both on the severity of the underlying disease and the severity of the MI, which may explain the poor prognosis in patients with Type 2 MI. Although higher age, often characteristic of Type 2 MI, is recognized as the cause of higher morality, in an investigation restricted to young adults (50 years and younger), Type 2 mortality was significantly higher than Type 1 mortality at 10.2 years (34.2% vs. 12%, respectively) [20]. In our study, the elevation of CPK in Type 2 MI patients was smaller than that for Type 1 MI patients, however, comparable prognoses were observed between Type 1 MI and Type 2 MI. When Type 2 MI was divided into two categories, the CPK was significantly higher in the Type 2 MI with Non-CVS group, which seemed to affect the poor prognosis compared with Type 2 MI with CVS. Type 2 MI was associated with fewer events and was adjusted for minimal factors, including age, Type 2 with Non-CVS remained a significant independent predictor for the mortality (Table 4). It has been reported that patients with Type 2 MI complicated by coronary arteriosclerosis have a worse prognosis than those without CAD [25,36], these arteriosclerotic patients may be included in the Type 2 MI with Non-CVS group of this study.

In the 3 Kaplan–Meier curves, Type 2 non-CVS mortality was found to be increased during the acute phase (Figure 2B). This feature is consistent with the prognosis of Type 2 MI in studies comparing the prognosis of Type 1 and Type 2 MI in the past [9,10,11,12]. The absence of this finding on the Type 2 pooled survival curve (Figure 2A) is characteristic of this study and cannot be compared in the absence of similar studies in the past, but it shows how the favorable prognosis of Type 2 with CVS had a significant impact on our results.

The frequency of CVS-related Type 2 MI is high in Japan [15,37] and about half of the Type 2 MI patients in the present study had CVS. It is well known that the incidence of coronary vasospastic angina is higher in Japanese than in Western patients [15]. Even though smoking is an important acquired factor related to the onset of CVS [38] and despite the higher rate of smoking among Japanese patients with AMI as compared with Western patients [39], the genetic background is considered to be associated with the onset of vasospastic angina [40,41,42].

The effectiveness of medication for coronary vasospastic angina has already been established. In this patient population, more CCBs and nitrates were administered to patients with Type 2 MI with CVS, which suggests that appropriate medical therapy is being performed in Japan. CCBs are considered a first-line drug for the prevention of coronary vasospastic angina. Nitrates are also a first-line drug for releasing the angina attack and compensates for the decline in nitric oxide activity due to impaired coronary endothelial function [43]. In spasming coronary arteries, nitrate has a higher vasodilatory activity than CCBs [44], effectively treating the spasm in a manner different to that of CCB. Thus, it is preferable to use nitrates and CCB selectively, according to the appropriate indication.

Thrombus formation is another important consideration for AMI onset due to CVS [30]. At the time of coronary spasm, coagulation accelerates, fibrinolytic activity decreases, and the activities of platelets and adhesion molecules increase. Thrombus formation also leads to the secretion of platelet-derived growth factors, which further exacerbate thrombus formation and can promote vasoconstriction [45,46,47]. Further studies are required to assess the efficacy of antiplatelet medications for coronary vasospastic angina for patients with Type 2 MI with CVS [48]. Given the high incidence of CVS among Japanese AMI patients, further risk stratification re: with/without CVS, together with medications and PCI, would optimize AMI outcomes for Japanese patients.

### Study Limitations

The present study has several limitations. First, this study was a post-hoc analysis and not a population-based study, which raises the possibility of selection bias. Second, no harmonized adjudication process was used to determine the diagnosis with respect to disease type classification during patient enrollment. Third, we did not collect information on the causes of Type 2 MI with non-CVS thus, the underlying disease(s) affecting patient prognosis other than MI and CVS are unknown. Fourth, only patients who were admitted within 48 hours of MI onset were enrolled. Therefore, patients who were admitted after 48 hours were excluded, and this may have contributed, in part, to the lower incidence of NSTEMI patients. Fifth, a diagnosis of coronary vasospastic angina is difficult and was left to the discretion of the attending physician at each facility. In the acute phase of AMI, coronary spasm may develop transiently, making a diagnosis more difficult. Sixth, in this study, some patients were diagnosed with MI by CPK. Furthermore, the concentration of cTn and the time interval data (delta change) were not prespecified in this study, which is not representative of current clinical diagnostic standards.

In this study, 793 patients (26.5%) completed 3-year follow-up. There is a possibility that the low follow-up rate of the registered patients may have caused some bias. We have additionally performed survival analyses at 1-year follow-up (follow up rate was 84.9%). The consistency of findings, compared with 3-year analysis, were confirmed on Kaplan–Meyer analysis. The event rates in each categorized group and the Kaplan–Meyer analysis are shown in the Appendix A.

## 5. Conclusions

The equivalent mortality rate observed between Type 1 and Type 2 MI among Japanese patients is attributable to a significantly lower mortality rate in patients with Type 2 MI with CVS.

## Figures and Tables

**Figure 1 jcm-09-01686-f001:**
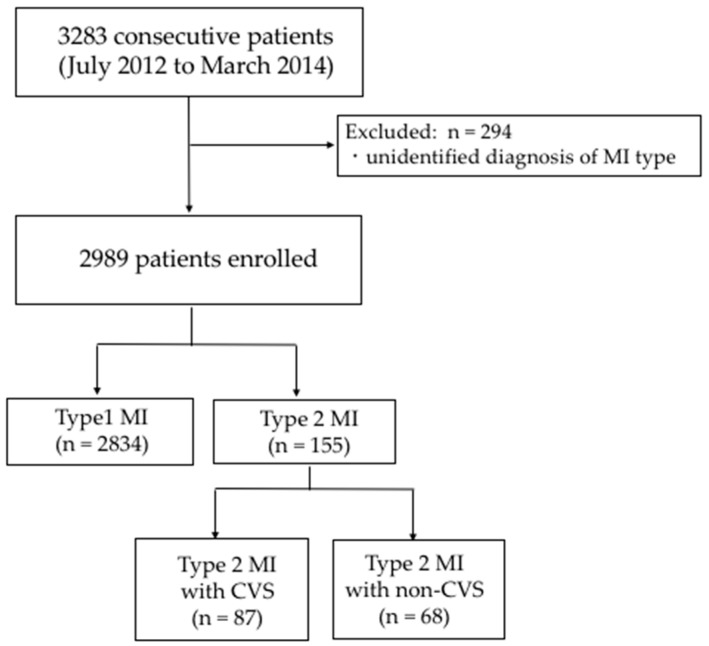
Flowchart of the current study. MI, myocardial infarction; CVS, coronary vasospasm.

**Figure 2 jcm-09-01686-f002:**
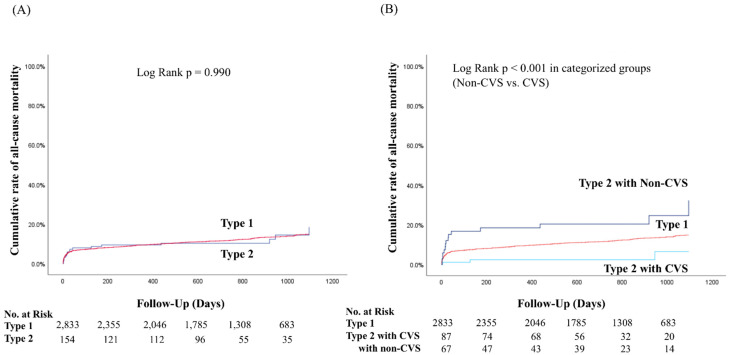
Kaplan–Meier estimates of all-cause mortality between Type 1 MI and Type 2 MI (**A**) and Table 2. MI is stratified into two groups (**B**) over a 3-year period.

**Figure 3 jcm-09-01686-f003:**
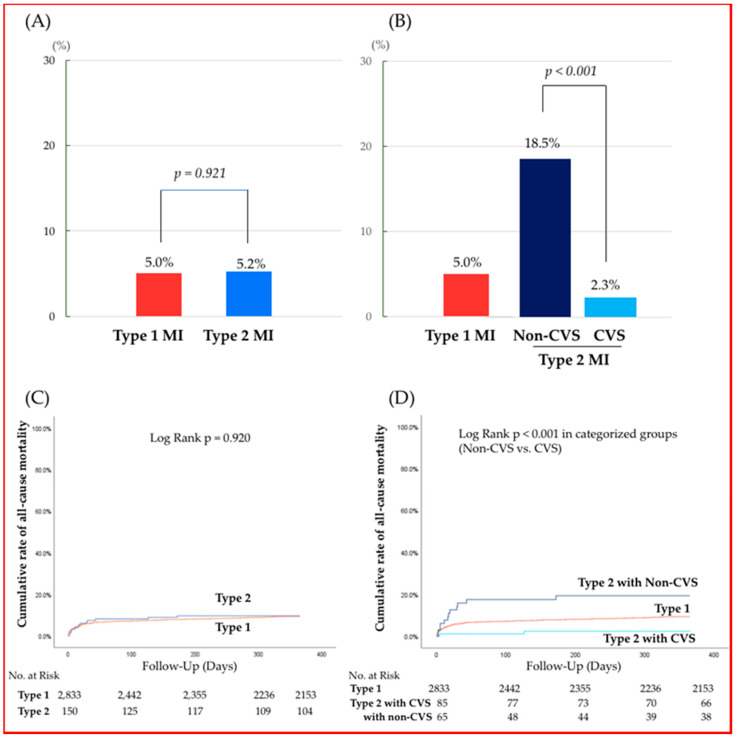
Event rates and Kaplan–Meier estimates of all-cause mortality at 1year period. (Upper) All-cause mortality between Type 1 MI and Type 2 MI (**A**) and the mortality among the 3 groups, when Type 2 MI is stratified into 2 groups (**B**) within 1-year period. (Lower) Kaplan–Meier estimates of all-cause mortality between Type 1 MI and Type 2 MI (**C**) and the mortality among the three groups, when Type 2 MI is stratified into two groups (**D**) within 1-year period.

**Table 1 jcm-09-01686-t001:** Baseline patient characteristics.

	Type 1 MI (*n* = 2834)	Type 2 MI (*n* = 155)	*p*	Type 2 MI		
				Non-CVS (*n* = 68)	With CVS (*n* = 87)	*p*
Age (years)	68.7 ± 12.5	65.8 ± 14.4	0.003	72.4 ± 11.9	60.7 ± 14.5	0.029
Female (%)	699 (24.7)	53 (34.2)	0.008	23 (33.8)	30 (34.5)	0.930
BMI (kg/m^2^)	23.6 ± 3.9	23.1 ± 3.2	0.156	22.7 ± 3.6	23.2 ± 3.9	0.112
Onset to admission (min)	345.7 ± 479.5	382.8 ± 553.3	0.062	337.6 ± 506.0	418.2 ± 588.2	0.381
HR at admission (bpm)	79.0 ± 21.7	78.9 ± 19.8	0.340	83.3 ± 22.6	75.1 ± 16.8	0.033
Killip Class (III/IV)	433 (15.3)	13 (8.4)	0.018	11 (16.2)	2 (2.3)	0.007
Hypertension (%)	1885 (67.0)	103 (66.5)	0.886	44 (64.7)	59 (67.8)	0.910
Diabetes mellitus (%)	1015 (36.6)	40 (25.8)	0.006	24 (35.3)	16 (18.4)	0.002
Dyslipidemia (%)	1435 (51.2)	95 (61.3)	0.015	43 (63.2)	52 (59.8)	0.047
Chronic kidney disease (%)	1261 (44.5)	68 (43.9)	0.876	43 (63.2)	25 (28.7)	<0.001
Current smoking (%)	921(33.6)	51 (33.8)	0.972	19 (27.9)	32 (38.6)	0.388
Prior MI (%)	350 (12.4)	18 (11.6)	0.769	11 (16.2)	7 (8.0)	0.299
Prior PCI (%)	424 (15.1)	32 (20.6)	0.062	24 (35.3)	8(9.2)	<0.001
Atrial fibrillation (%)	166 (5.6)	12 (7.8)	0.347	9 (13.4)	3 (3.4)	0.023
Prior stroke (%)	276 (9.9)	17 (10.9)	0.651	12 (17.6)	5 (5.8)	0.041
Composite of PAD (%)	121 (4.5)	14 (9.4)	0.006	11 (17.2)	3 (3.5)	<0.001
Urgent CAG (%)	2655 (93.8)	130 (83.9)	<0.001	59 (86.8)	71 (81.6)	<0.001
Max CPK (IU/L)	2444.1 ± 2952.8	1065.4 ± 1828.3	<0.001	1712.5 ± 2347.7	559.7 ± 1049.9	<0.001
Hemoglobin (g/dL)	13.7 ± 3.3	12.9 ± 2.5	0.009	11.7 ± 2.9	13.9 ± 1.7	<0.001
eGFR (mL/min/1.73 m^2^)	65.8 ± 45.7	63.6 ± 27.8	0.006	50.7 ± 27.3	73.7 ± 24.0	0.006
LDL-cholesterol (mg/dL)	117.5 ± 37.8	100.8 ± 37.4	<0.001	95.5 ± 37.1	104.5 ± 37.4	<0.001
HDL-cholesterol (mg/dL)	46.7 ± 13.4	49.8 ± 15.2	0.006	46.2 ± 14.8	52.6 ± 15.1	0.003
HbA1c (%)	6.3 ± 1.4	5.9 ± 0.8	<0.001	6.1 ± 0.8	5.9 ± 0.7	<0.001
NSTEMI (%)	837 (29.5)	93 (60.0)	<0.001	34 (50.0)	59 (67.8)	<0.001
Urgent revascularization (%)	2,485 (87.8)	63 (40.6)	<0.001	42 (61.8)	21 (24.4)	<0.001

Data are mean ± standard deviation (SD), median (interquartile range), or percentage. BMI, body mass index; HR, heart rate; MI, myocardial infarction; PCI, percutaneous coronary intervention; PAD, peripheral artery disease; CAG, coronary angiography; PAD, peripheral artery disease; CPK, creatine phosphokinase; eGFR, estimated glomerular filtration rate; LDL, low-density lipoprotein; HDL, high-density lipoprotein; NSTEMI, non-ST elevation myocardial infarction.

**Table 2 jcm-09-01686-t002:** Medication treatments at discharge.

	Type 1 MI (*n* = 2834)	Type 2 MI (*n* = 155)	*p*	Type 2 MI		
				Non-CVS (*n* = 68)	With CVS (*n* = 87)	*p*
Aspirin (%)	2562 (97.3)	70 (45.1)	<0.001	39 (70.9)	54 (65.9)	0.530
DAPT (%)	2058 (78.2)	52 (33.5)	<0.001	27 (49.1)	25 (30.5)	<0.001
Anticoagulant Therapy (%)	327 (12.4)	24 (15.5)	0.131	21 (36.2)	3 (3.5)	<0.001
CCBs (%)	551 (20.8)	94 (60.6)	<0.001	16 (27.6)	78 (90.7)	<0.001
β-blockades (%)	1838 (70.2)	53 (34.2)	<0.001	37 (64.9)	16 (21.6)	<0.001
Nitrates (%)	246 (9.5)	37 (23.9)	<0.001	7 (12.7)	30 (38.5)	<0.001
Nicorandil (%)	541 (21.0)	41 (26.4)	<0.014	12 (21.8)	29 (36.3)	0.005
ACE-Is (%)	1364 (52.8)	52 (33.5)	0.003	28 (51.9)	24 (30.8)	0.001
ARBs (%)	739 (28.5)	41 (26.4)	0.679	18 (31.6)	23 (29.1)	0.874
Diuretics (%)	23 (0.9)	0 (0)	NA	0 (0)	0 (0)	>0.990
Statins (%)	2303 (87.6)	112 (72.3)	0.005	37 (67.3)	75 (87.2)	<0.001
Hypoglycemic Agents (%)	728 (27.5)	24 (15.5)	0.040	16 (27.6)	8 (9.3)	0.001
PPIs (%)	2260 (85.5)	92 (59.3)	<0.001	43 (74.1)	49 (57.0)	<0.001

DAPT, double antiplatelet therapy; CCB, calcium channel blocker; ACE-I, angiotensin; converting enzyme inhibitor; ARB, angiotensin II receptor blocker; PPI, proton pump inhibitor.

**Table 3 jcm-09-01686-t003:** Multiple regression analyses for predicting all-cause mortality in the Type 1 MI Group.

Variable	HR	95% CI	*P*
Age	1.043	1.026–1.060	<0.001
Female	1.270	0.899–1.794	0.175
BMI	0.978	0.939–1.019	0.978
Killip Class	1.986	1.750–2.254	<0.001
Current Smoking	1.061	0.731–1.540	0.756
Chronic Kidney Disease	2.298	1.611–3.276	<0.001
Prior PCI	1.664	1.158–2.393	0.006
Composite of PAD	2.045	1.203–3.475	0.008
Urgent Revascularization	0.737	0.441–1.230	0.243
Multivessel Disease	1.365	1.004–1.858	0.047

Abbreviations are shown in Table 1 and Table 2, HR, hazard ratio; CI, confidence interval.

**Table 4 jcm-09-01686-t004:** Multiple regression analyses for predicting all-cause mortality in the Type 2 MI Group.

Variable	HR	95% CI	*P*
Age	1.080	0.996–1.171	0.061
Heart Rate	1.040	0.999–1.081	0.053
Killip Class	0.814	0.355–1.871	0.629
Chronic Kidney Disease	1.144	0.242–5.410	0.865
Prior CABG	1.511	0.080–28.567	0.783
Urgent Revascularization	5.085	1.018–25.390	0.047
Multivessel Disease	5.395	1.186–24.545	0.029
Hemoglobin	0.992	0.712–1.384	0.964

Abbreviations are shown in Table 1, Table 2 and Table 3.

**Table 5 jcm-09-01686-t005:** Cox hazards model analysis in the type 2 MI adjusting for age, sex, and BMI.

Variable	HR	95% CI	*P*
Type 2 MI with CVS	Reference		
Type 2 MI with Non-CVS	5.007	1.338–18.741	0.017
Age	1.020	0.973–1.070	0.407
Sex	1.527	0.560–4.165	0.408
BMI	0.993	0.862–1.145	0.925

Abbreviations are shown in Table 1, Table 2 and Table 3.

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
