# Peer review of "Long-Term Prognosis of Patients with Myocardial Infarction Type 1 and Type 2 with and without Involvement of Coronary Vasospasm"

_jcm, 2020, doi:10.3390/jcm9061686_

Round 1
Reviewer 1 Report
No new comments
Author Response
Response to Reviewer 1 Comment
We thank the Reviewer for the careful review and comments. We are grateful for your evaluation.
Comments and Suggestion for authors:
No new comments
Response:
We are happy to receive a good review for our changes. Thank you very much.
Reviewer 2 Report
I agree with the editor's comment #8 that there is room to improve statistical analysis. The limited sample size may limit extensive control for all variables, but can the authors at least try to control for age, gender, BMI in their survival analyses? This analysis even without graphic presentation would be helpful for the readers.
Author Response
Please see the attachment.

This manuscript is a resubmission of an earlier submission. The following is a list of the peer review reports and author responses from that submission.
Round 1
Reviewer 1 Report
In the present paper Sato et al describes a multicenter, prospective observation of 2989 Japanese patients diagnosed with type 1 or type 2 myocardial infarction within 48 hours of hospital presentation. The authors investigated the occurrence of the primary outcome of all-cause mortality over a 3-year period. The major finding of the study was that the 3-year mortality was comparable in the type 1 (N=2834) and type 2 MI patients (N=155). When the type 2 MI patients were separated into 2 distinct groups based on the presence of coronary vasospasm on invasive testing, those with vasospasm (N=87) had a significantly lower 3-year mortality (3.4%) compared to the group without vasospasm (N=68, mortality: 3.4%). This is a potentially very interesting and novel finding, however there are many confounders that make the interpretation very difficult. The paper is overall well written and data is well presented.
- The patients with type 2 MI with vasospasm were significantly younger, had lower rate of diabetes, hyperlipidemia, CKD, prior PCI, atrial fibrillation, stroke, PAD and had higher hemoglobin and lower CPK levels. This suggests that this is a much lower risk population which can potentially explain the lower mortality rates. Was vasospasm entered in the Cox regression models? It would be important to see whether vasospasm is an independent predictor of mortality.
- Please provide data about the cardiac troponin levels. How often was CPK used for the diagnosis of MI in patients with vasospasm? This is important as CPK is not as specific for myocardial infarction. The 4th universal definition of myocardial infarction document recommends using troponin for the diagnosis of MI whenever possible and only use CK-MB otherwise.
- How was the date of death determined? In the introduction a separate section should be dedicated to outcomes.
- The literature suggests that most deaths in the type 2 MI group can be attributed to non-cardiac causes. Is there any information available about the cause of death? Is there a different cardiovascular and non-cardiovascular death rate in type 2 MI with or without vasospasm?
- How was the vasospasm defined? Please provide detailed description about the provocation test used (dose, coronaries tested), What was the percent response threshold to consider the testing positive?
- The proportion of type 2 MI is particularly low in this study? What are the potential contributors?
Minor points:
- Was there any loss to follow-up? This should be clearly stated in the manuscript.
- In Table 1 there are 2 lines describing the patients’ age with different numbers.
- Clinical outcomes and definitions: technically the diagnosis of MI was probably based on the third universal definition of MI as enrollment started on 2012, please cite the appropriate citation.
Reviewer 2 Report
In the present manuscript Sato et al report their findings on the comparison of type 2 (vasospasm-related) to type 1 myocardial infarction. My comments on the manuscript are:
- Introduction: the more recent studies comparing Type 1 and Type 2 MI (e.g. APACE study, BACC study and doi.org/10.1016/j.jacc.2019.12.052) should be referenced and mentioned.
- The hypothesis should be written more precicly to state, that the authors aim to compare the outcome of type 2 MI patients with and without vasospasm.
- Methods: There is more information on the adjudication process required. How exactly was the adjudication performed? Was is done by two physicians in a blinded fashion? The authors state, that the new UDMI was used. Does this refer to the 4th UDMI? If so, which sex-specific cutoffs and which delta of troponin were used (which algorithm)? Which troponin assay was used?
- Was myocardial injury documented as a separate category? If so, this specific outcome should also be reported.
- What were the causes for Type 2 MI, except from vasospasm?
- How was the follow-up performed? How were patients contacted? How many patients were lost to follow-up? What was the median follow-up time?
- More information in the cause of death is required. How many patients died from CVD or cancer?
- I am worried about using CPK to diagnose MI, as this does not represent clinical standard anymore. Thus, I would suggest to exclude these patients.
- The conclusion is hard to understand – I would recommend to remove the addition “because of the lower mortality rate of type 2 MI with CVS”.
- Table 1: What does the p-value indicate? Please add information on baseline and serial troponin.
Round 2
Reviewer 1 Report
The authors have answered my questions, only 2 minor requests/questions remain.
Point 2 - Please include the number of patients diagnosed with CK in the manuscript text. The provided figure does not need to be included.
Point 6 - What do you mean by "sophisticated institutions"? This section needs to be revised. How were these conditions excluded and why? Was there any oversight for inclusion/exclusion?
Please reword: "Furthermore, sophisticated institutions in Japan were selected as registered institutions for this study. Therefore, it is possible that Type 2 MI other than coronary artery disease, such as Takotsubo cardiomyopathy or pulmonary embolism, which are strictly classified as Type 2 MI (categorized as myocardial injury), were strictly excluded at the time of registration."
Reviewer 2 Report
Thank you for providing the updated manuscript, which addresses several of my comments. So specific comments remain:
- The authors have added some new manuscripts to the introduction, but some of the most important papers are still missing, which include e.g. 10.1161/CIRCULATIONAHA.119.043100 and http://dx.doi.org/10.1016/j.jacc.2017.07.774
- Adjudication process: The more detailed information provided in the response letter should also be included in the manuscript. I am confused by the comment “adjudication for MI type and prognosis should have been determined by two or more experienced physicians at each facility”. Was this predefined or are you just expecting this? In case the latter is true, you should mention in the limitations section, that no harmonized adjudication process was used to determine the diagnosis. The lack of a specific delta change should also be acknowledged.
- Regarding the information on the duration and completeness of follow-up, the detailed information provided in the response letter should also be included in the manuscript.
- I still am worried about using CPK to diagnose MI, as this does not represent clinical standard anymore. Thus, I would again suggest to exclude these patients.
- Baseline table: Are you able to add the baseline troponin concentrations?
